# Cumulative ROC curves for discriminating three or more ordinal outcomes with cutpoints on a shared continuous measurement scale

**B. Rey deCastro**¤*

National Center for Environmental Health, Centers for Disease Control and Prevention, Atlanta, Georgia, United States of America

¤ Current address: National Center for HIV/AIDS, Viral Hepatitis, STD, and TB Prevention, Centers for Disease Control and Prevention, Atlanta, Georgia, United States of America
* rdecastro@cdc.gov

**Data Availability Statement:** All relevant data are within the manuscript and its Supporting Information files.

**Funding:** The author received no specific funding for this work.

## Abstract

Cumulative receiver operator characteristic (ROC) curve analysis extends classic ROC curve analysis to discriminate three or more ordinal outcome levels on a shared continuous scale. The procedure combines cumulative logit regression with a cumulative extension to the ROC curve and performs as expected with ternary (three-level) ordinal outcomes under a variety of simulated conditions (unbalanced data, proportional and non-proportional odds, areas under the ROC curve [AUCs] from 0.70 to 0.95). Simulations also compared several criteria for selecting cutpoints to discriminate outcome levels: the Youden Index, Matthews Correlation Coefficient, Total Accuracy, and Markedness. Total Accuracy demonstrated the least absolute percent-bias. Cutpoints computed from maximum likelihood regression parameters demonstrated bias that was often negligible. The procedure was also applied to publicly available data related to computer imaging and biomarker exposure science, yielding good to excellent AUCs, as well as cutpoints with sensitivities and specificities of commensurate quality. Implementation of cumulative ROC curve analysis and extension to more than three outcome levels are straightforward. The author's programs for ternary ordinal outcomes are publicly available.

## Introduction

Classic receiver operator characteristic (ROC) curve analysis addresses the relation of continuous measurements to binary outcomes [1], and enables selection of a cutpoint or threshold on the continuous measurement scale discriminating the outcome levels. From its origins in signal detection theory [2] and application in early radio detection and ranging systems, the technique has been used in fields as diverse as clinical chemistry [3], radiology [4], psychology [5], and machine learning [6–9].

**Competing interests:** The author has declared that no competing interests exist.

Extension beyond binary outcomes would be desirable for the increased scope of applications. One readily implemented approach is to group multinomial outcome levels into binomial levels and run classic ROC curve analysis, but this loses information and biases test accuracy [10]. There have been other, more sophisticated proposals spanning a range of theoretical approaches [11–18], but the complexity of these noteworthy proposals has limited their application. Additional methods have been implemented and some enjoy broad use [19–22], yet theoretical justification may be sparse.

This paper proposes a two-stage, semiparametric approach combining conventional cumulative logit regression with a cumulative extension of ROC curve analysis to discriminate ordinal outcome levels. The performance of this approach is evaluated under simulation, with comparison of several criteria used with classic ROC curves to select cutpoints. Results from these criteria are compared to cutpoints computed from maximum likelihood estimates (MLEs) of the regression parameters. The procedure is also demonstrated with publicly available data.

## Formulation

The classic empirical ROC curve is computed by comparing a binary outcome $Y$ with a continuous measure $X$ where each observed level of $X$ is evaluated as a candidate cutpoint discriminating observed $Y = 1$ (positive) from $Y = 2$ (negative). Observations exceeding the candidate cutpoint are classified positive with respect to the continuous measurement, while those less than or equal to the cutpoint are classified negative. As in a $2 \times 2$ contingency table, the count of correct classifications among positive outcomes comprises the true positives ($TP$) and among negative outcomes the true negatives ($TN$). The count of incorrect classifications among negative outcomes comprises the false positives ($FP$) and among positive outcomes the false negatives ($FN$). These counts are used to compute: sensitivity, which is the probability that an observation with a positive outcome is correctly classified by a continuous measurement above a candidate cutpoint (sensitivity = TP/[TP+FN]); and specificity, which is the probability that an observed negative outcome is correctly classified by a continuous measurement at or below a candidate cutpoint (specificity = TN/[TN+FP]). Thence, coordinates for the empirical ROC curve are computed where the abscissa is 1 − specificity (= false positive rate; FPR) and the ordinate is sensitivity (= true positive rate; TPR).

The best cutpoint $X^*$ given the data may be identified from the ROC curve coordinates with a criterion that maximizes TPR and minimizes FPR. Cross-referencing the identified ROC curve coordinate with its observed continuous measurement yields the cutpoint distinguishing the binary outcomes. A variety of cutpoint criteria are available, such as the Youden Index, Matthews Correlation Coefficient, and Total Accuracy [23–25]. In addition, the ability of the continuous measurement to discriminate between outcome levels, which is equivalent to the strength of the association between the two, may be represented by the area under the ROC curve (AUC; also known as the c-statistic), which is the probability that an observation with a positive outcome will have a higher continuous measurement than an observation with a negative outcome.

Since the ROC curve describes the relationship between a binary outcome and continuous predictor, it is directly related to logistic regression [26–28]. For the binary outcome where $Y = 2$ is the reference outcome level, let $\pi_1 = \Pr[Y = 1]$. The univariate logistic model with continuous predictor $X$ and linear parameters $\{\alpha, \beta\}$ is:

$$\text{logit}\,(\pi_1) = \log\left(\frac{\pi_1}{1 - \pi_1}\right) = \alpha + \beta X \tag{1}$$

Let $\hat{\pi}_{1i}$ be the probability that $Y = 1$ predicted by the regression model at the $i$th observation $X_i$ for $i = 1, \ldots, N$ where $N$ is the number of observations. Analogous to the approach above, each predicted probability may serve as a candidate cutpoint discriminating $Y = 1$ from $Y = 2$. Coordinates comprising the ROC curve may then be computed, except they are based on counts on the probability scale monotonically transformed by the regression model from the original continuous scale. As above, the best probability cutpoint $\hat{\pi}^*$ may be selected with a suitable optimality criterion. The cutpoint on the scale of the continuous predictor $X^*$ can be recovered by cross-referencing $\hat{\pi}^*$ with its corresponding observed measurement. Whether using the scale of the continuous predictor or the predicted probability, the resulting ROC curves are identical because the curve is rank-based and invariant to monotonic transformations of the continuous predictor [1]. The first stage of the proposed approach exploits this invariance through a generalization of the logistic model embodied by the cumulative logit model.

## Stage 1: Cumulative logit model

The cumulative logit regression model predicts probabilities for an ordinal outcome $Y = j$ with $j = 1, \ldots, J$ levels, where for demonstration $J = 3$ and the reference outcome level is $Y = 3$. Let $\pi_j = \Pr[Y \leq j]$, then with continuous predictor $X$ and linear parameters $\{\alpha_j, \beta_j\}$ the cumulative logit regression model is:

$$\text{logit}\,(\pi_j) = \log\left(\frac{\pi_j}{1 - \pi_j}\right) = \alpha_j + \beta_j X, \ \text{for}\ j = 1, \ldots, J-1 \tag{2}$$

For each $j$th outcome level up to $J - 1$, a cumulative logit is estimated with its own regression parameters $\{\hat{\alpha}_j, \hat{\beta}_j\}$. The formulation in Eq 2 is known as non-proportional odds where the log-odds $\hat{\beta}_j$ differs between outcome levels. Simplification is possible with the proportional odds formulation where constant log-odds $\beta_j = \beta$ is assumed among outcome levels [29]. Let $\hat{\pi}_{ij} = \hat{\Pr}[Y_{ij} \leq j | X_i]$, then in terms of the parameter estimates $\{\hat{\alpha}_j, \hat{\beta}_j\}$ the predicted cumulative probability for the $j$th outcome level at the $i$th observation is:

$$\hat{\pi}_{ij} = \frac{\exp\left(\hat{\alpha}_j + \hat{\beta}_j X_i\right)}{1 + \exp\left(\hat{\alpha}_j + \hat{\beta}_j X_i\right)}, \quad \text{for}\ j = 1, \ldots, J-1 \tag{3}$$

and the predicted individual probability is the difference of adjacent cumulative probabilities:

$$\frac{\exp\left(\hat{\alpha}_j + \hat{\beta}_j X_i\right)}{1 + \exp\left(\hat{\alpha}_j + \hat{\beta}_j X_i\right)} - \frac{\exp\left(\hat{\alpha}_{j-1} + \hat{\beta}_{j-1} X_i\right)}{1 + \exp\left(\hat{\alpha}_{j-1} + \hat{\beta}_{j-1} X_i\right)} \tag{4}$$

For $J = 2$ this model reduces to the logistic model, but the cumulative logit model is similarly able to transform the continuous predictor to the predicted probability scale except that each outcome level gets a predicted probability function.

We have so far recalled that ROC curves are invariant to monotonic transformation of the continuous measurement, including transformation by a logistic regression model to the predicted probability scale. In addition, we have reviewed the cumulative logit model and its transformation of a single continuous predictor to a series of separate predicted probabilities for each level of the ordinal outcome. These probabilities are comprehensive and mutually

exclusive with respect to the outcome $\left( \sum_j \hat{\pi}_{ij} = 1 \right)$ and are suitable for computing a series of "cumulative" ROC curves.

## Stage 2: Cumulative ROC curves

Calculation of the classic ROC curve on the predicted probability scale can be readily extended to count *TP*, *TN*, *FP*, and *FN* for each cumulative logit, resulting in $J - 1$ cumulative ROC curves. For the cumulative logit associated with the *j*th outcome, let $p_{jk}$ be the *k*th candidate cumulative probability cutpoint from among the $\hat{\pi}_{ij}$, then one may count $TP_{jk}$, $TN_{jk}$, $FP_{jk}$, and $FN_{jk}$ with the indicator function $I(\cdot)$ by comparing the outcome $Y_i$ with $\hat{\pi}_{ij}$ vs. $p_{jk}$ for $i = 1, \ldots, N$; $j = 1, \ldots, J - 1$; and $k = 1, \ldots, N$:

$$TP_{jk} = \sum_i I\left(\hat{\pi}_{ij} > p_{jk} \quad \text{AND} \quad Y_i \leq j\right) \tag{5}$$

$$TN_{jk} = \sum_i I\left(\hat{\pi}_{ij} \leq p_{jk} \quad \text{AND} \quad Y_i > j\right) \tag{6}$$

$$FP_{jk} = \sum_i I\left(\hat{\pi}_{ij} > p_{jk} \quad \text{AND} \quad Y_i > j\right) \tag{7}$$

$$FN_{jk} = \sum_i I\left(\hat{\pi}_{ij} \leq p_{jk} \quad \text{AND} \quad Y_i \leq j\right) \tag{8}$$

From these counts, the coordinates ($FPR_{jk}$, $TPR_{jk}$) for the *j*th cumulative $ROC_j$ curve can be computed, where $FPR_{jk} = 1 - (TN_{jk}/[TN_{jk} + FP_{jk}])$ and $TPR_{jk} = (TP_{jk}/[TP_{jk} + FN_{jk}])$. Continuing with the case of the ternary ordinal outcome, $p_{1k}$ is the *k*th candidate cutpoint from the first cumulative logit and $p_{2k}$ from the second, so that the cumulative $ROC_1$ curve discriminates between $Y = 1$ vs. $Y = 2$ or 3, and the cumulative $ROC_2$ curve discriminates between $Y = 1$ or 2 vs. $Y = 3$. Analogous to the binary case, a probability cutpoint for the *j*th outcome level $\hat{\pi}_j^*$ may be selected from its respective cumulative $ROC_j$ curve using a suitable criterion. The cutpoint on the scale of the continuous predictor $X_j^*$ is recovered by cross-referencing $\hat{\pi}_j^*$ with its corresponding observed measurement.

Alternatively, ROC curve analysis may be forgone altogether by computing cutpoints from the MLE cumulative logit regression parameters, where $X_j^* = -\left(\hat{\alpha}_j / \hat{\beta}_j\right)$. Since this parametric cutpoint is the ratio of two model parameters, both the Delta Method and Fieller's Method are applicable for computing the variance [30]. Fieller's Method is favored, however, since it tends to provide better coverage despite potential asymmetry of the confidence interval [31, 32]. In addition, Hirschberg and Lye 2010 [31] recommend Fieller's Method when the computed ratios are positive and correlation between the numerator and denominator is negative. Accordingly, the standard deviation $s_{X^*}$ is estimated here with Fieller's Method [33] when computing confidence intervals for parametric cutpoints as $\pm \left(t_{df = 2, 1-(\alpha/2)} \times s_{X^*}\right)$.

## Simulations

Cumulative ROC curve analysis for a ternary ordinal outcome was evaluated under conditions simulating AUCs = 0.70, 0.75, 0.85, 0.90, and 0.95. Cutpoints for the continuous predictor were set at $X_2^* = -5$ and $X_3^* = 5$ by designating $\alpha_j$ and $\beta_j$ based on the relationship $X_j^* = -(\alpha_j / \beta_j)$. Random variates of the continuous predictor were obtained from a normal distribution $X_i \sim N$

**Table 1. Cutpoint selection criteria based on evaluation of empirical ROC curves.**

| Criterion | Formula | Range |
|---|---|---|
| Youden Index (or Informedness, $\Delta P'$) | $sensitivity + specificity - 1$ | (0,1) |
| Matthews Correlation Coefficient | $\frac{(TP \times TN) - (FP \times FN)}{([TP+FP][TP+FN][TN+FP][TN+FN])^{1/2}}$ | (−1,1) |
| Total Accuracy | $\frac{TP+TN}{TP+FN+TN+FP}$ | (0,1) |
| Markedness ($\Delta P$) | $\frac{TP}{TP+FP} + \frac{TN}{TN+FN} - 1$ | (0,1) |

(0, $\sigma^2 = 100$) truncated at the 10th and 90th percentiles. Truncation improved the chances of obtaining random variates that would successfully converge to a maximum likelihood solution for the regression model. Random variates of the ternary outcome $Y_i$ were then obtained from a multinomial distribution defined by probabilities computed from Eq 4 with $\alpha_j$, $\beta_j$, and random variates $X_i$. For the proportional odds condition with $AUC_1 = AUC_2 = 0.90$, parameters were designated at $\alpha_1 = -1.70$, $\alpha_2 = 1.70$, and $\beta = 0.34$. For the first non-proportional odds condition (referred to as the NPO1 condition) with $AUC_1 = 0.75$ and $AUC_2 = 0.85$, parameters were designated at $\alpha_1 = -0.75$, $\beta_1 = 0.15$ and $\alpha_2 = 1.25$, $\beta_2 = 0.25$; and for the second (NPO2 condition) with $AUC_1 = 0.70$ and $AUC_2 = 0.95$, parameters were $\alpha_1 = -0.70$, $\beta_1 = 0.14$ and $\alpha_2 = 4.70$, $\beta_2 = 0.94$. For each condition, 10,000 datasets were simulated with nested sample sizes $n = 75$, 150, and 300 unequally allocated among the outcome levels. A cumulative logit regression model was fit to each dataset and cumulative ROC curves computed. Simulations were run with the FREQ, LOGISTIC, and SURVEYSELECT subroutines of the SAS® software application, version 9.4 [34].

Several cutpoint selection criteria were evaluated for their ability to correctly identify designated cutpoints from cumulative ROC curves: the Youden Index (also known as Informedness and $\Delta P'$), Matthews Correlation Coefficient, Total Accuracy, and Markedness ($\Delta P$) [35, 36]. These criteria and their ranges are presented in Table 1. Each criterion embodies certain merits, but all achieve their optimal level at the ROC curve coordinate where the criterion is at its observed maximum. Cutpoints were also computed directly from MLE cumulative logit regression parameters.

Tables 2 and 3 confirm that distributions realized during the proportional odds and NPO1 simulations were approximately centered at the levels designated above for $\alpha_j$, $\beta_j$, and $AUC$. Table 4 shows, however, that for the NPO2 condition the medians of the realized distributions for $\alpha_j$ and $\beta_j$ were about 14–30 percent above designated levels for $\alpha_1$, $\alpha_2$, and $\beta_2$ and about 14 percent below for $\beta_1$. Designated levels for all regression parameters, however, were between the 2.5th and 97.5th percentiles of their realized distributions, and the realized AUCs were centered on their designated values.

**Table 2. Proportional odds simulation.** Parameter estimates and AUCs realized from cumulative ROC curve analysis of 10,000 simulated datasets parameterized with proportional odds and $AUC_1 = AUC_2 = 0.90$.

| | | | | | |
|---|---|---|---|---|---|
| **$\alpha_1 = -1.70$, $\alpha_2 = 1.70$, $\beta = 0.34$, $AUC_1 = AUC_2 = 0.90$** | | | | | |
| **n** | **$\hat{\alpha}_1$** [2.5th, 97.5th %ile] | **$\hat{\alpha}_2$** [2.5th, 97.5th %ile] | **$\hat{\beta}$** [2.5th, 97.5th %ile] | **$AUC_1$** [2.5th, 97.5th %ile] | **$AUC_2$** [2.5th, 97.5th %ile] |
| 75 | −1.745 [−2.749, −1.048] | 1.742 [1.034, 2.728] | 0.352 [0.248, 0.509] | 0.9004 [0.8119, 0.9631] | 0.8997 [0.8112, 0.9632] |
| 150 | −1.724 [−2.343, −1.230] | 1.723 [1.236, 2.346] | 0.346 [0.272, 0.443] | 0.8982 [0.8403, 0.9447] | 0.8978 [0.8404, 0.9452] |
| 300 | −1.713 [−2.127, −1.355] | 1.713 [1.357, 2.127] | 0.343 [0.290, 0.408] | 0.8974 [0.8576, 0.9317] | 0.8975 [0.8568, 0.9322] |

**Table 3. Non-proportional odds simulation: NPO1.** Parameter estimates and AUCs realized from cumulative ROC curve analysis applied to 10,000 simulated datasets parameterized for non-proportional odds, $AUC_1 = 0.75$, and $AUC_2 = 0.85$.

$\alpha_1 = -0.75, \alpha_2 = 1.25, \beta_1 = 0.15, \beta_2 = 0.25, AUC_1 = 0.75, AUC_2 = 0.85$

| n | $\hat{\alpha}_1$ [2.5th, 97.5th %ile] | $\hat{\alpha}_2$ [2.5th, 97.5th %ile] | $\hat{\beta}_1$ [2.5th, 97.5th %ile] | $\hat{\beta}_2$ [2.5th, 97.5th %ile] | $AUC_1$ [2.5th, 97.5th %ile] | $AUC_2$ [2.5th, 97.5th %ile] |
|---|---|---|---|---|---|---|
| 75 | −0.759 [−1.421, −0.226] | 1.284 [0.656, 2.289] | 0.153 [0.065, 0.267] | 0.259 [0.154, 0.451] | 0.7492 [0.6158, 0.8605] | 0.8445 [0.7416, 0.9247] |
| 150 | −0.758 [−1.179, −0.392] | 1.267 [0.817, 1.859] | 0.152 [0.092, 0.224] | 0.253 [0.181, 0.364] | 0.7491 [0.6623, 0.8265] | 0.8458 [0.7758, 0.9059] |
| 300 | −0.755 [−1.042, −0.493] | 1.260 [0.935, 1.641] | 0.151 [0.108, 0.200] | 0.252 [0.198, 0.318] | 0.7490 [0.6886, 0.8051] | 0.8463 [0.7961, 0.8898] |

Tables 5–7 display the median, 2.5th and 97.5th percentiles, and percent-bias of cutpoints selected by each criterion across sample sizes *n*. Percent-biases are the median of percent-differences between realized cutpoints (selected and parametric) and designated cutpoints. The ROC curve-based cutpoint selection criteria exhibited a range of biases. Among both proportional (Table 5) and non-proportional (Tables 6 and 7) odds conditions, absolute values of the percent-biases ranged from 2.8–144.2 percent. Total Accuracy demonstrated the best performance with biases ranging from 2.8–11.7 percent, while the other criteria performed considerably worse.

Forgoing ROC curve analysis and calculating cutpoints from the MLE regression parameters yielded small, often negligible absolute percent-biases (<2.3 percent) for both proportional and non-proportional odds conditions. In addition, across all sample sizes, parametric cutpoints consistently out-performed ROC curve-based cutpoint selection criteria.

Notably, for the NPO2 condition (Table 7), divergence of the realized cumulative logit parameters from designated values did not entail discrepancies in realized cutpoints compared to the other simulation conditions, whether the cutpoints were selected by criteria or calculated parametrically.

Absolute percent-bias for Total Accuracy cutpoints usually diminished with increasing sample size from *n* = 75 to 300. The only exception was for the upper cutpoint (*X* = 5.00) in the NPO2 simulation condition, where absolute percent-bias worsened slightly at *n* = 150. In addition, the absolute percent-bias of lower Total Accuracy cutpoints were usually greater than for upper cutpoints, except for *n* = 150 and 300 of the NPO2 condition. For parametric cutpoints, absolute percent-bias was negligible at all sample sizes, except in the NPO2 condition, where although cutpoint bias was the lowest within the condition (0.1–2.3 percent), it was slightly greater compared to other conditions (0.0–0.3 percent).

**Table 4. Non-proportional odds simulation: NPO2.** Parameter estimates and AUCs realized from cumulative ROC curve analysis applied to 10,000 simulated datasets parameterized for non-proportional odds, $AUC_1 = 0.70$, and $AUC_2 = 0.95$.

$\alpha_1 = -0.70, \alpha_2 = 4.70, \beta_1 = 0.14, \beta_2 = 0.94, AUC_1 = 0.70, AUC_2 = 0.95$

| n | $\hat{\alpha}_1$ [2.5th, 97.5th %ile] | $\hat{\alpha}_2$ [2.5th, 97.5th %ile] | $\hat{\beta}_1$ [2.5th, 97.5th %ile] | $\hat{\beta}_2$ [2.5th, 97.5th %ile] | $AUC_1$ [2.5th, 97.5th %ile] | $AUC_2$ [2.5th, 97.5th %ile] |
|---|---|---|---|---|---|---|
| 75 | −0.501 [−1.209, 0.103] | 6.071 [3.243, 20.611] | 0.098 [−0.008, 0.230] | 1.229 [0.669, 4.370] | 0.6979 [0.4786, 0.8292] | 0.9483 [0.8488, 0.9872] |
| 150 | −0.499 [−0.967, −0.079] | 5.768 [3.648, 14.259] | 0.098 [0.022, 0.183] | 1.150 [0.741, 2.959] | 0.6941 [0.5906, 0.7870] | 0.9505 [0.8934, 0.9847] |
| 300 | −0.520 [−0.832, −0.217] | 5.426 [3.943, 8.782] | 0.101 [0.048, 0.159] | 1.069 [0.791, 1.763] | 0.6931 [0.6233, 0.7616] | 0.9508 [0.9148, 0.9769] |

**Table 5. Proportional odds simulation.** Percentiles and biases of cutpoints selected from cumulative ROC curves with several criteria and computed parametrically: 10,000 simulated datasets parameterized for proportional odds and $AUC_1 = AUC_2 = 0.90$.

| Cutpoint | Criterion | n = 75 | | n = 150 | | n = 300 | |
|---|---|---|---|---|---|---|---|
| | | %Bias | Median [2.5th, 97.5th %ile] | %Bias | Median [2.5th, 97.5th %ile] | %Bias | Median [2.5th, 97.5th %ile] |
| 5.00 | Youden Index | −46.4 | 2.68 [−0.71, 6.00] | −48.5 | 2.58 [−0.28, 5.22] | −50.4 | 2.48 [0.26, 4.62] |
| | Total Accuracy | 7.7 | 5.39 [1.88, 9.24] | 4.8 | 5.24 [2.48, 8.16] | 2.9 | 5.15 [2.95, 7.38] |
| | Matthews Correlation | −18.8 | 4.06 [−0.06, 8.22] | −19.9 | 4.01 [0.59, 7.29] | −22.1 | 3.89 [1.14, 6.58] |
| | Markedness | 78.7 | 8.93 [3.05, 12.51] | 99.4 | 9.97 [4.35, 12.64] | 120.5 | 11.02 [5.58, 12.73] |
| | Parametric $-(\hat{\alpha}_j/\hat{\beta}_j)$ | −0.2 | 4.99 [3.06, 7.22] | −0.1 | 5.00 [3.62, 6.51] | −0.1 | 5.00 [4.01, 6.06] |
| −5.00 | Youden Index | 52.0 | −2.40 [−5.65, 1.08] | 52.1 | −2.39 [−5.06, 0.30] | 52.0 | −2.40 [−4.51, −0.24] |
| | Total Accuracy | 10.0 | −4.50 [−8.22, −1.09] | 5.9 | −4.70 [−7.66, −2.02] | 4.1 | −4.79 [−7.05, −2.64] |
| | Matthews Correlation | 24.6 | −3.77 [−7.79, 0.36] | 23.6 | −3.82 [−7.11, −0.43] | 23.8 | −3.81 [−6.46, −1.04] |
| | Markedness | −70.8 | −8.54 [−12.05, −2.60] | −95.2 | −9.76 [−12.40, −4.22] | −117.9 | −10.89 [−12.60, −5.61] |
| | Parametric $-(\hat{\alpha}_j/\hat{\beta}_j)$ | 0.2 | −4.99 [−7.23, −3.05] | −0.1 | −5.00 [−6.50, −3.64] | 0.2 | −4.99 [−6.04, −4.03] |

**Table 6. Non-proportional odds simulation: NPO1.** Percentiles and biases of cutpoints selected from cumulative ROC curves with several criteria and computed parametrically: 10,000 simulated datasets parameterized for non-proportional odds, $AUC_1 = 0.75$, and $AUC_2 = 0.85$.

| Cutpoint | Criterion | n = 75 | | n = 150 | | n = 300 | |
|---|---|---|---|---|---|---|---|
| | | %Bias | Median [2.5th, 97.5th %ile] | %Bias | Median [2.5th, 97.5th %ile] | %Bias | Median [2.5th, 97.5th %ile] |
| 5.00 | Youden Index | −74.3 | 1.29 [−3.92, 6.41] | −77.8 | 1.11 [−3.17, 5.40] | −79.9 | 1 [−2.45, 4.38] |
| | Total Accuracy | 8.4 | 5.42 [−0.40, 11.64] | 5.3 | 5.26 [0.58, 10.47] | 3.5 | 5.17 [1.43, 9.19] |
| | Matthews Correlation | −51.2 | 2.44 [−4.99, 9.73] | −56.2 | 2.19 [−4.08, 8.39] | −58.7 | 2.06 [−3.11, 7.07] |
| | Markedness | 113.7 | 10.69 [−8.21, 12.73] | 133.6 | 11.68 [−7.40, 12.78] | 144.2 | 12.21 [−0.81, 12.80] |
| | Parametric $-(\hat{\alpha}_j/\hat{\beta}_j)$ | 0.1 | 5 [1.47, 11.81] | −0.2 | 4.99 [2.50, 8.76] | 0.0 | 5.00 [3.21, 7.37] |
| −5.00 | Youden Index | 67.3 | −1.63 [−5.60, 2.43] | 65.1 | −1.75 [−4.94, 1.39] | 64.7 | −1.77 [−4.35, 0.76] |
| | Total Accuracy | 11.7 | −4.41 [−9.03, −0.32] | 7.0 | −4.65 [−8.19, −1.39] | 4.4 | −4.78 [−7.52, −2.15] |
| | Matthews Correlation | 38.3 | −3.08 [−8.21, 2.14] | 35.7 | −3.22 [−7.34, 1.06] | 36.3 | −3.18 [−6.58, 0.32] |
| | Markedness | −85.7 | −9.28 [−12.27, −1.38] | −116.5 | −10.83 [−12.57, −4.31] | −134.6 | −11.73 [−12.70, −6.29] |
| | Parametric $-(\hat{\alpha}_j/\hat{\beta}_j)$ | 0.3 | −4.99 [−8.08, −2.60] | −0.1 | −5.01 [−7.03, −3.30] | 0.3 | −4.99 [−6.35, −3.82] |

**Table 7. Non-proportional odds simulation: NPO2.** Percentiles and biases of cutpoints selected from cumulative ROC curves with several criteria and computed para-metrically: 10,000 simulated datasets parameterized for non-proportional odds, $AUC_1 = 0.70$, and $AUC_2 = 0.95$.

| Cutpoint | Criterion | n = 75 | | n = 150 | | n = 300 | |
|---|---|---|---|---|---|---|---|
| | | %Bias | Median [2.5th, 97.5th %ile] | %Bias | Median [2.5th, 97.5th %ile] | %Bias | Median [2.5th, 97.5th %ile] |
| 5.00 | Youden Index | −42.6 | 2.87 [−2.16, 8.43] | −43.4 | 2.83 [−1.30, 7.12] | −45.5 | 2.73 [−0.66, 6.25] |
| | Total Accuracy | 4.6 | 5.23 [−2.74, 11.55] | 5.0 | 5.25 | 3.3 | 5.17 [1.28, 9.38] |
| | Matthews Correlation | −28.3 | 3.59 [−3.47, 10.90] | −30.3 | 3.48 [−2.89, 10.14] | −32.9 | 3.36 [−1.83, 8.74] |
| | Markedness | 105.2 | 10.26 [−5.38, 12.72] | 131.8 | 11.59 [−5.76, 12.78] | 143.9 | 12.2 [−6.08, 12.80] |
| | Parametric $-(\hat{\alpha}_j/\hat{\beta}_j)$ | −1.4 | 4.93 [−8.12, 30.83] | 1.5 | 5.07 [0.94, 14.99] | 2.3 | 5.11 [2.55, 9.54] |
| −5.00 | Youden Index | 41.4 | −2.93 [−4.77, −0.28] | 44.1 | −2.79 [−4.46, −0.59] | 46.4 | −2.68 [−4.04, −0.96] |
| | Total Accuracy | 9.4 | −4.53 [−6.17, −2.73] | 5.0 | −4.75 [−6.07, −3.38] | 2.8 | −4.86 [−5.98, −3.75] |
| | Matthews Correlation | 24.5 | −3.78 [−5.77, −1.37] | 20.1 | −3.99 [−5.58, −2.09] | 19.7 | −4.02 [−5.31, −2.45] |
| | Markedness | 2.3 | −4.88 [−6.55, −2.06] | −8.0 | −5.4 [−6.59, −3.43] | −16.0 | −5.8 [−6.63, −4.35] |
| | Parametric $-(\hat{\alpha}_j/\hat{\beta}_j)$ | 1.5 | −4.93 [−5.96, −3.76] | −0.1 | −5.00 [−5.75, −4.20] | −1.3 | −5.06 [−5.59, −4.51] |

## Real-World Data

Two publicly available datasets with ternary ordinal outcomes were analyzed with the cumulative ROC curve approach where cutpoints were selected with the Total Accuracy criterion and computed parametrically. Fig 1 displays histograms for each dataset overlaid with Total Accuracy cutpoints, while Fig 2 shows the cutpoints on their respective cumulative ROC curves.

Tables 8 and 9 present the Total Accuracy and parametric cutpoints, as well as their sensitivities, specificities, and AUCs. Confidence intervals for parametric cutpoints were calculated with Fieller's Method [33] and for AUCs with Wald's Method [37]. Cumulative logit regression models and cumulative ROC curves were computed with the FREQ and LOGISTIC subroutines of the SAS software application, version 9.4 [34].

### Cork stopper quality

The data comprise measurements of material defects appearing in digital images of cork stoppers [38, 39], available in S1 File. An automated image processing system scanned cork defects and quantified several characteristics, including the number, area, and perimeter of the defects. Fifty cork stoppers were quantified in each of three quality levels subjectively assigned by human experts (N = 150), where Y = 1 (poor), 2 (normal), and 3 (superior). In the Stage 1 cumulative logit model, cork stopper quality was predicted by the total number of pixels with defects [px]. The score test for proportional odds (p-value = 0.31) supported a proportional odds configuration for the model. The parameter estimates are: $\hat{\alpha}_1 = -13.64$, $\hat{\alpha}_2 = -7.05$, and $\hat{\beta}_{area} = -0.036$.

The ability of defect area to discriminate cork stopper quality is excellent, with AUCs > 0.97 for both cumulative ROC curves (Table 8). Total Accuracy identified cutpoints where the

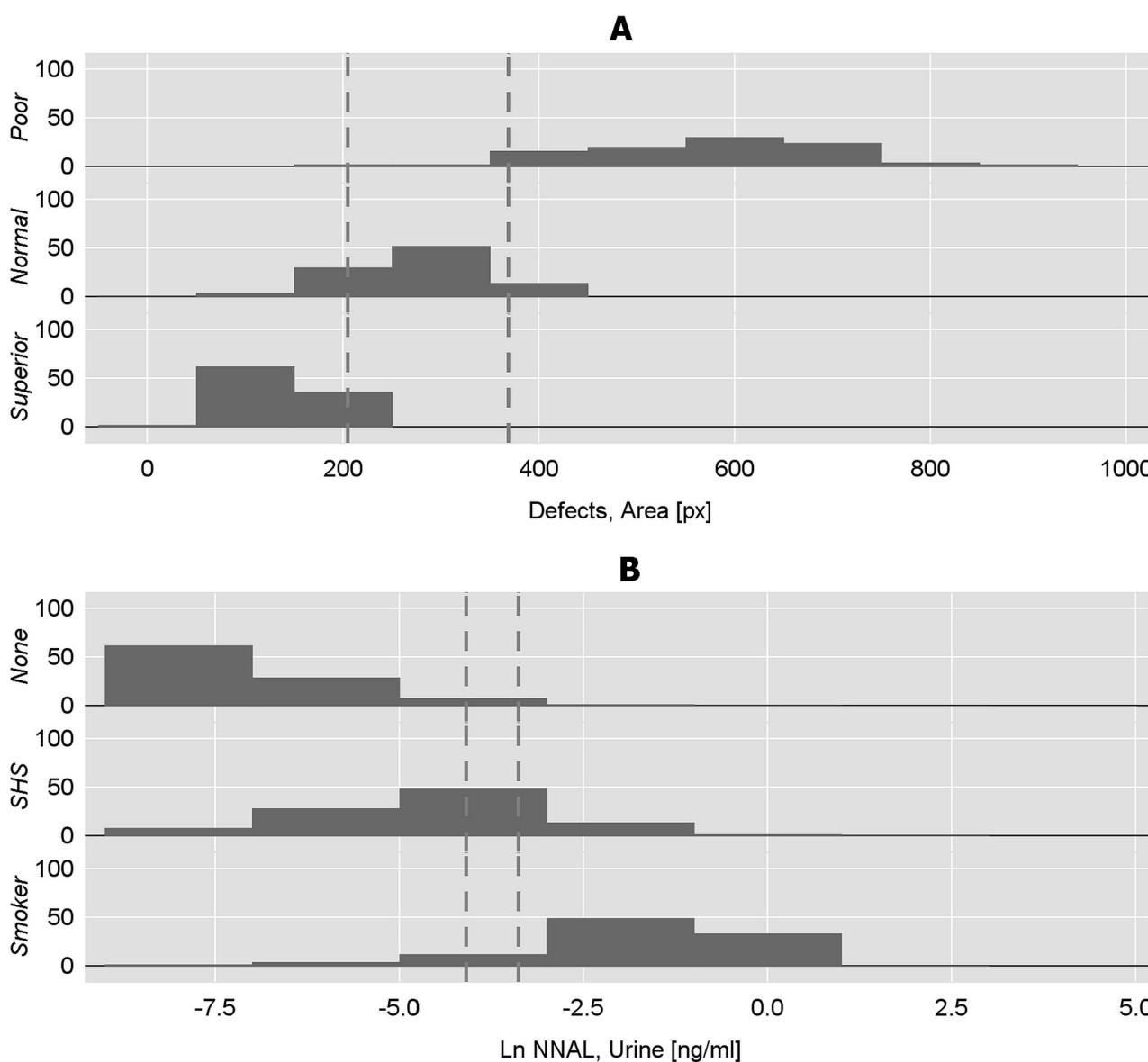

**Fig 1. Percent distributions overlaid with cutpoints (dashed lines) selected from cumulative ROC curves with the Total Accuracy criterion.** A: Cork Stopper Quality (N = 150): total defective area [px] by cork stopper quality levels. B: NHANES Tobacco Smoke Exposure (N = 16,900): natural log of urinary NNAL [ng/mL] by tobacco smoke exposure levels.

total number of pixels with defects were 205 (distinguishing poor or normal quality vs. superior) and 369 (poor vs. normal, superior), and both had excellent sensitivities and specificities > 0.93. Parametrically computed cutpoints were at 194.8 [95%CI: 177.1, 213.7] and 376.8 [352.9, 403.9] pixels, with sensitivities and specificities comparable to those of the Total Accuracy cutpoints, although specificity for the lower cutpoint and sensitivity for the upper cutpoint were somewhat attenuated.

## NHANES tobacco smoke exposure

Human exposure to chemicals can be estimated from measurements of trace compounds in samples of human urine. Some of these compounds, known as biomarkers, are associated with

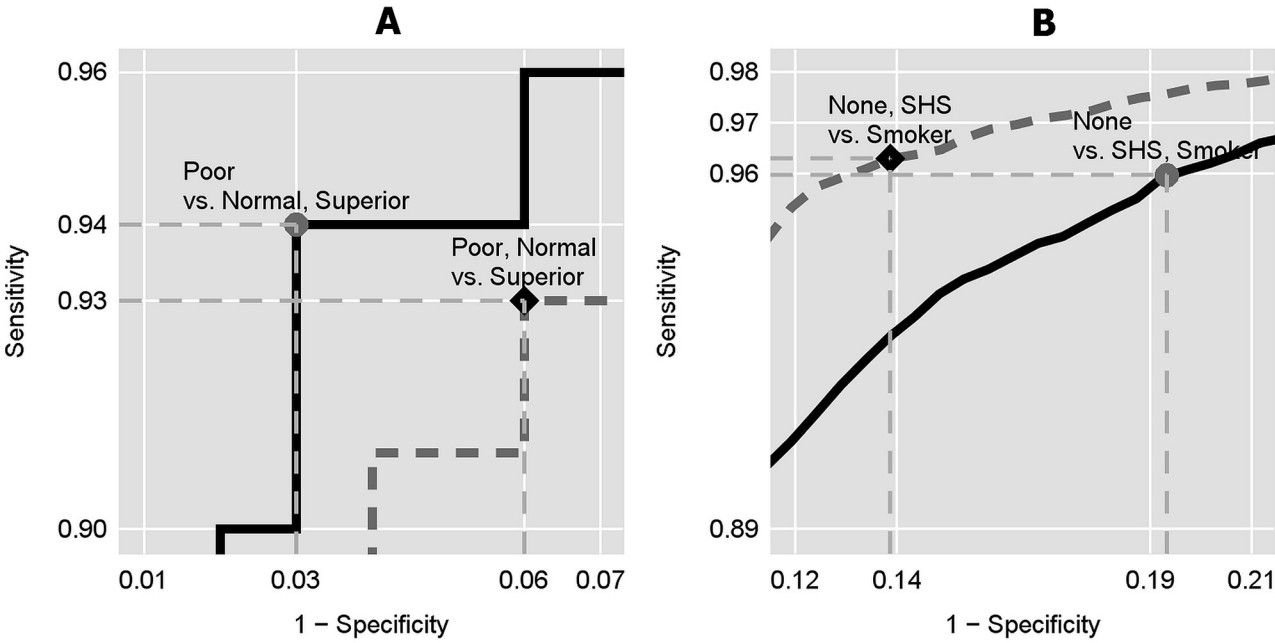

**Fig 2. Cumulative ROC curves (focused on upper-left quadrant) indicating coordinates for Total Accuracy cutpoints.** A: Cork Stopper Quality: poor vs. normal, superior quality (solid line) and poor, normal vs. superior (dashed line). B: NHANES Tobacco Smoke Exposure: none vs. second-hand smoke (SHS), smoker (solid line) and none, SHS vs. smoker (dashed line).

exposure to tobacco smoke, which may arise either from direct inhalation while smoking, or from indirect inhalation of tobacco smoke present in the environment (i.e., second-hand tobacco smoke; SHS). One such biomarker is a tobacco-specific N-nitrosamine known as NNAL (4-[methylnitrosamino]-1-[3-pyridyl]-1-butanol; CAS No. 76014-81-8), which is present in both mainstream tobacco smoke and smokeless tobacco products. NNAL was measured in urine from a representative multi-level sample of the United States civilian population $\geq 12$ years old ($N = 16, 900$) obtained during the 2007–2012 cycles of the National Health and Nutrition Examination Survey (NHANES) [40], available in S2 File. Subjects reported being in one of three ordinal exposure categories: non-exposed subjects who neither used tobacco

**Table 8. Cork stopper quality (N = 150).** Cutpoints of total defective area [px] discriminating cork stopper quality levels were selected from cumulative ROC curves with Total Accuracy criterion and computed parametrically. Proportional odds were assumed among quality levels.

| Categories | Cutpoint | Sn: Sensitivity Sp: Specificity | AUC [95%CI] |
|---|---|---|---|
| **Total Accuracy** | | | |
| Poor vs. | 369 | Sn: 0.9400 | 0.9847 |
| Normal, Superior | | Sp: 0.9700 | [0.9655, 1.0000] |
| Poor, Normal vs. | 205 | Sn: 0.9300 | 0.9732 |
| Superior | | Sp: 0.9400 | [0.9501, 0.9963] |
| **Parametric [95CI]** | | | |
| Poor vs. | 376.8 | Sn: 0.8800 | — |
| Normal, Superior | [352.9, 403.9] | Sp: 0.9800 | |
| Poor, Normal vs. | 194.8 | Sn: 0.9300 | — |
| Superior | [177.1, 213.7] | Sp: 0.9000 | |

**Table 9. Tobacco smoke exposure (N = 16,990).** Cutpoints of the natural log of urinary NNAL [ng/ml] discriminating tobacco smoke exposure levels. Cutpoints were selected from cumulative ROC curves with Total Accuracy criterion and computed parametrically. Non-proportional odds were assumed among exposure levels.

| Categories | Cutpoint | Sn: Sensitivity Sp: Specificity | AUC [95%CI] |
|---|---|---|---|
| **Total Accuracy** | | | |
| Non-Exposed vs. | −4.092 | Sn: 0.9597 | 0.9535 |
| SHS, Exclusive Smokers | | Sp: 0.8060 | [0.9497, 0.9573] |
| Non-Exposed, SHS vs. | −3.168 | Sn: 0.9689 | 0.9679 |
| Exclusive Smokers | | Sp: 0.8458 | [0.9646, 0.9712] |
| **Parametric [95CI]** | | | |
| Non-Exposed vs. | −4.053 | Sn: 0.8029 | — |
| SHS, Exclusive Smokers | [−4.108, −3.998] | Sp: 0.9602 | |
| Non-Exposed, SHS vs. | −3.264 | Sn: 0.8537 | — |
| Exclusive Smokers | [−3.319, −3.208] | Sp: 0.9661 | |

products nor were exposed to SHS ($Y = 1; n_1 = 12, 372$); SHS-exposed subjects who did not smoke tobacco ($Y = 2; n_2 = 927$); and exclusive tobacco smokers ($Y = 3; n_3 = 3, 691$). In order to eliminate a non-combustible source of NNAL, subjects were excluded from analysis if they reported using smokeless tobacco. The natural log of urinary NNAL concentration predicted self-reported exposure categories in the Stage 1 cumulative logit model. The score test ($p$-value <0.001) supported a non-proportional odds configuration for the model. The parameter estimates are: $\hat{\alpha}_1 = -4.60$, $\hat{\alpha}_2 = -4.08$, $\hat{\beta}_{ln(NNAL),1} = -1.13$, and $\hat{\beta}_{ln(NNAL),2} = -1.25$.

The ability of $ln(NNAL)$ to discriminate ternary tobacco smoke exposure levels is excellent with AUCs > 0.95 for both cumulative ROC curves (Table 9). Total Accuracy identified cutpoints at $ln(NNAL)$ concentrations of -4.092 (non-exposed vs. SHS-exposed, smokers) and -3.168 ng/mL (non-exposed, SHS-exposed vs. smokers). Exponentiated, the respective cutpoints are 16.71 and 42.09 pg/mL. Since the non-proportional odds configuration permits each cumulative ROC curve to differ in discriminatory power, the cumulative ROC curve associated with the lower cutpoint had an AUC of 0.9535 [95%CI: 0.9497, 0.9573], while the upper cutpoint's curve had an AUC that was slightly, but significantly better at 0.9679 [0.9646, 0.9712]. Parametric cutpoints are at −4.053 [95%CI: −4.108, −3.998] and −3.264 [−3.319, −3.208] ng/mL (exponentiated: 17.37 [16.44, 18.35] and 38.24 [36.19, 40.44] pg/mL, respectively). Total Accuracy's upper cutpoint was above the parametric upper cutpoint's upper 95 percent confidence limit, but it is unclear which is preferable. The Total Accuracy upper cutpoint had excellent sensitivity (0.9689) and good specificity (0.8458), but this was reversed for the parametric upper cutpoint, which had good sensitivity (0.8537) and excellent specificity (0.9661). Another basis for comparison is the Total Accuracy criterion, which can be calculated for parametric cutpoints from their $TP_j$, $TN_j$, $FP_j$, and $FN_j$. This, too, failed to be conclusive since the criterion for the Total Accuracy vs. parametric upper cutpoints were hardly different at 0.9421 vs. 0.9417.

## Discussion

The cumulative logit model subsumes multinomial ordinal outcome levels within a single model, yet each outcome level gets its own cumulative logit, so that predicted individual probabilities for each level (Eq 4) are mutually exclusive, comprehensive over the outcome levels, and sum to unity for each observation of the continuous measurement. Another appeal of the model is that its predicted probabilities (Eqs 3 and 4) change in direct proportion to the

continuous measurement across all outcome levels. Even more, the ordinality of the outcome ensures that cutpoints separate successive pairs of adjacent outcome levels.

The assumption of proportional odds constrains the log-odds of the continuous predictor to be constant for all levels of the ordinal outcome. This imposes statistical equivalence on the AUCs of the cumulative ROC curves, so that the ROC curves will appear approximately overlapped. In contrast, when the log-odds of the predictor are non-proportional, which represents varying strength in the continuous predictor's association at each outcome level, the AUCs of the cumulative ROC curves will differ and the curves will appear nested. Notably, the rank-order of the AUCs (and hence the order of nesting) is independent of the order of the ordinal outcome levels. This flexibility may be especially desirable in certain settings, such as in a clinical trial where a medication may be associated with greater potency at the worst level of the health outcome.

Evaluated under simulation, cumulative ROC curve analysis performed as expected for a variety of conditions, but with the qualification that if ROC curve-based cutpoint criteria are to be used, results from simulated unbalanced data indicate that Total Accuracy yields minimally biased cutpoints compared to the Youden Index, Matthews Correlation Coefficient, and Markedness. In contrast to cutpoints selected by criteria, parametric cutpoints have the advantage of being maximum likelihood and consequently had absolute percent-biases that were less than Total Accuracy's and were often negligible.

The previously noted divergence of the cumulative logit parameters in the NPO2 simulation condition also suggests that caution may be warranted in some non-proportional odds situations, particularly when AUCs of the cumulative ROC curves are widely separated, as in NPO2. If, however, the primary aim is cutpoint estimation, the NPO2 condition indicates that estimated cutpoints were robust against divergence in the logit parameters. Moreover, qualitative results from the NPO2 condition regarding cutpoint selection criteria and parametric cutpoints were consistent with those from the proportional odds and NPO1 simulations.

Analysis of real-world data demonstrated that cumulative ROC curve analysis yields reasonable results. Continuous measurements in both datasets displayed varying degrees of overlap among the ternary outcome levels. The tobacco smoke exposure data were relatively large, but also strikingly unbalanced across the outcome levels, especially at the intermediate outcome level. The intermediate SHS-exposed category was small (5.5 percent) and the distribution was skewed toward the extreme exposure levels (72.8 percent non-exposed vs. 21.7 percent smokers). Notwithstanding, the cumulative ROC curve approach identified cutpoints with good to excellent sensitivity and specificity.

The cumulative ROC curve approach readily generalizes to more than three outcome levels through specification of the cumulative logit model. Nonetheless, discriminating discrete outcome levels postulates that the continuous measurement is associated with an *a priori* number of latent and ordinal classes. If the cumulative logit model in Stage 1 specifies an outcome with $J > 2$ ordinal levels, determination of cutpoints may be difficult if the outcome is actually binomial or is otherwise different than assumed. The magnitude of this difficulty may be revealed in exploratory data analysis, by poor model fit, and by cutpoints with poor sensitivity and specificity. For the tobacco smoke exposure data, although prior assumption of an intermediate outcome level (i.e., secondhand smoke-exposed) was plausible, there was cause for doubt since this level was observed infrequently. In addition, the infrequency of the SHS-exposed outcome level contrasts with the simulated data, where use of a normal distribution as a source of random variates for the continuous predictor leads to more frequent intermediate outcomes (~40 percent) compared to the extreme outcomes (~20 percent each). These considerations notwithstanding, the natural log of urinary NNAL was an excellent discriminator of the three tobacco smoke exposure levels.

The proposed approach admits alternative formulations of the Stage 1 model, where other multinomial models may be implemented through substitution of the cumulative logit link function. Alternative models for ordinal outcomes, such as adjacent categories and continuation ratio (including complementary log-log and Cox proportional hazards), and nominal outcomes (with the generalized logit) all predict probabilities entirely suitable for subsequent calculation of cumulative ROC curves. Conceptual interpretation of these alternative link functions, however, necessarily varies, sometimes substantially, and may therefore be less directly interpretable than the cumulative logit. Exploring the performance and utility of these alternative link functions may nonetheless be fruitful.

Cumulative ROC curve analysis appears to be efficacious for a univariate continuous predictor, and the regression framework may be extended with the addition of covariates to the Stage 1 cumulative logit model. This can be expected to enhance discriminatory power by accounting for other influential or potentially confounding influences [41]. In any particular case, however, it may not be clear whether additional covariates will adversely affect the overall concavity of the cumulative ROC curves for the continuous predictor of interest, thereby hindering selection of cutpoints. Stratification by discrete factors may be helpful in resolving some of these difficulties.

One challenge posed by the cumulative logit model is its sample size demands, which arise from the potentially numerous parameters that must be estimated. In the proportional odds configuration, the univariate cumulative logit model has $J - 1$ intercepts plus one slope for the continuous predictor, but this nearly doubles in the non-proportional odds configuration, which has $2 \times (J - 1)$ regression parameters.

## Conclusions

The cumulative ROC curve method comprises a straightforward combination of cumulative logit regression with ROC curve analysis, and is readily implemented with available statistical software. Cutpoint selection criteria from classic ROC curve analysis are still applicable, as well as established performance measures, such as sensitivity, specificity, and AUC. Cumulative ROC curve analysis performed as expected under simulation and with real-world data for a variety of conditions, including balanced and unbalanced data, proportional and non-proportional odds assumptions for the cumulative logit model, and AUCs associated with fair, good, and excellent performance (AUC = 0.70–0.95). Of the ROC curve-based cutpoint criteria, Total Accuracy was the least biased in simulation compared to the Youden Index, Mathews Correlation Coefficient, and Markedness. Calculation of cutpoints from cumulative logit regression parameters, which forgoes evaluation of cumulative ROC curves, demonstrated minimal bias, owing to parameter estimation with maximum likelihood methods. The author's SAS programs implementing cumulative ROC curve analysis for ternary ordinal outcomes ($J = 3$) with parametric cutpoints are freely available for download in S1 Programs and from the author's GitHub repository [42]: https://github.com/intelligo1466/cumRoc3.

## Supporting information

**S1 Programs. %cumRoc3—Cumulative ROC curve analysis of three-level ordinal outcomes.** A SAS macro that implements cumulative ROC curve analysis for three-level (ternary) ordinal outcomes, as described in this article. Requires SAS v9.4 or later.
(ZIP)

**S1 File. Data, cork quality.** Demonstration dataset comprising a ternary ordinal outcome representing levels of cork quality and a predictor representing the number of image pixels

exhibiting defects.
(ZIP)

**S2 File. Data, NHANES NNAL tobacco smoke exposure.** Demonstration dataset comprising a ternary ordinal outcome representing levels of self-reported tobacco smoke exposure and a predictor representing measurements of a tobacco-specific biomarker in urine.
(ZIP)

## Acknowledgments

The author is indebted to this journal's associate editors and reviewers for their gracious and constructive comments that significantly improved this paper. **Disclaimers**: The findings and conclusions in this report are those of the author and do not necessarily represent the views of the Centers for Disease Control and Prevention. Use of trade names is for identification only and does not imply endorsement by the Centers for Disease Control and Prevention.

## Author Contributions

**Conceptualization:** B. Rey deCastro.

**Data curation:** B. Rey deCastro.

**Formal analysis:** B. Rey deCastro.

**Investigation:** B. Rey deCastro.

**Methodology:** B. Rey deCastro.

**Software:** B. Rey deCastro.

**Validation:** B. Rey deCastro.

**Visualization:** B. Rey deCastro.

**Writing – original draft:** B. Rey deCastro.

**Writing – review & editing:** B. Rey deCastro.

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
