## [Decision Letter · Decision Letter 0]

4 Jul 2019

PONE-D-19-14500

Cumulative ROC curves for discriminating three or more ordinal outcomes with cutpoints on a shared continuous measurement scale

PLOS ONE

Dear Dr. deCastro,

Thank you for submitting your manuscript to PLOS ONE. After careful consideration, we feel that it has merit but does not fully meet PLOS ONE’s publication criteria as it currently stands. Therefore, we invite you to submit a revised version of the manuscript that addresses the points raised during the review process.

We would appreciate receiving your revised manuscript by Aug 18 2019 11:59PM. To enhance the reproducibility of your results, we recommend that if applicable you deposit your laboratory protocols in protocols.io, where a protocol can be assigned its own identifier (DOI) such that it can be cited independently in the future. For instructions see: http://journals.plos.org/plosone/s/submission-guidelines#loc-laboratory-protocols

We look forward to receiving your revised manuscript.

Kind regards,

Alessandro Parolari, MD, PhD

Academic Editor

PLOS ONE

**Journal Requirements:**

**Comments to the Author**

1. Is the manuscript technically sound, and do the data support the conclusions?

Reviewer #1: Yes

Reviewer #2: Yes

2. Has the statistical analysis been performed appropriately and rigorously? 

Reviewer #1: Yes

Reviewer #2: Yes

3. Have the authors made all data underlying the findings in their manuscript fully available?

Reviewer #1: Yes

Reviewer #2: Yes

4. Is the manuscript presented in an intelligible fashion and written in standard English?

Reviewer #1: Yes

Reviewer #2: Yes

5. Review Comments to the Author

Reviewer #1: The paper is well written, even if the subject is rather complex. The methodology is sound and the conclusions are in line with the results obtained. I have a few minor comments and modifications to suggest:

1. Line 192: it would be helpful to specify on which base (arbitrary or objective?) the three quality levels for the cork stoppers were established.

2. The discussion of the simulations is not deep enough (lines 268-274). For instance, the author should discuss the contrasting results of NPO1 and NPO2 (tables 4), where a substantial under- or over-estimation of the parameters occurr .

3. the Discussion includes some methodological specification that should be moved to a previous section: for example, 321-327 should be moved before the ‘Simulations’ section.

4. The same for the last two sentences of the Conclusions, which contain methodological specifications

Reviewer #2: I read with interest the manuscript. It's a very statistical summary of the employment of cumulative ROC curves for discriminating ordinal outcomes. I have no major comment

6. PLOS authors have the option to publish the peer review history of their article (what does this mean?). If published, this will include your full peer review and any attached files.

Reviewer #1: No

Reviewer #2: No

---

## [Author Response · Author response to Decision Letter 0]

11 Jul 2019

Please see uploaded PONE-D-19-14500_R1_Response to Reviewers.pdf

---

## [Editor Report · Decision Letter 1]

7 Aug 2019

Cumulative ROC curves for discriminating three or more ordinal outcomes with cutpoints on a shared continuous measurement scale

PONE-D-19-14500R1

Dear Dr. deCastro,

We are pleased to inform you that your manuscript has been judged scientifically suitable for publication and will be formally accepted for publication once it complies with all outstanding technical requirements.

With kind regards,

Alessandro Parolari, MD, PhD

Academic Editor

PLOS ONE
---

## [Editor Report · Acceptance letter]

9 Aug 2019

PONE-D-19-14918R1 

Cumulative ROC curves for discriminating three or more ordinal outcomes with cutpoints on a shared continuous measurement scale 

Dear Dr. deCastro:

I am pleased to inform you that your manuscript has been deemed suitable for publication in PLOS ONE. Congratulations! Your manuscript is now with our production department. 

With kind regards,

on behalf of

Dr Alessandro Parolari 

Academic Editor

PLOS ONE